# SARS-CoV-2 transmission in students of public schools of Catalonia (Spain) after a month of reopening

Anna Llupià[1,2,3]*, Alícia Borràs-Santos[4], Caterina Guinovart[2], Mireia Utzet[5,6,7], David Moriña[8], Joaquim Puig[9]

**1** Department of Preventive Medicine and Epidemiology, Hospital Clínic, Barcelona, Spain, **2** Barcelona Institute for Global Health (ISGlobal), Hospital Clínic-Universitat de Barcelona, Barcelona, Spain, **3** School of Medicine, Universitat de Barcelona, Barcelona, Spain, **4** Gimbernat School of Nursing, Universitat Autònoma de Barcelona (UAB), Sant Cugat del Vallès, Barcelona, Spain, **5** Center for Research in Occupational Health (CISAL), Barcelona, Spain, **6** Universitat Pompeu Fabra, CIBER of Epidemiology and Public Health (CIBERESP), Barcelona, Spain, **7** IMIM (Hospital del Mar Medical Research Institute), Barcelona, Spain, **8** Department of Econometrics, Statistics and Applied Econometrics, Riskcenter-IREA, Universitat de Barcelona, Barcelona, Spain, **9** Department of Mathematics, Universitat Politècnica de Catalunya, Barcelona, Spain

\* allupia@clinic.cat

**Data Availability Statement:** All relevant data are within the manuscript and its Supporting information files.

## Abstract

### Introduction

SARS-CoV-2 transmission within schools and its contribution to community transmission are still a matter of debate.

### Methods

A retrospective cohort study in all public schools in Catalonia was conducted using publicly available data assessing the association between the number of reported SARS-CoV-2 cases among students and staff in weeks 1–2 (Sept 14-27th, 2020) of the academic year with school SARS-CoV-2 incidence among students in weeks 4–5. A multilevel Poisson regression model adjusted for the community incidence in the corresponding basic health area (BHA) and the type of school (primary or secondary), with random effects at the sanitary region and BHA levels, was performed.

### Results

A total of 2184 public schools opened on September 14th with 778,715 students. Multivariate analysis showed a significant association between the total number of SARS-CoV-2 cases in a centre in weeks 1–2 and the SARS-CoV-2 school incidence among students in weeks 4–5 (Risk Ratio (RR) 1.074, 95% CI 1.044–1.105, p-value <0.001). The adjusted BHA incidence in the first two weeks was associated with school incidence in weeks 4–5 (RR 1.002, 95% CI 1.002–1.003, p-value <0.001). Secondary schools showed an increased incidence in weeks 4 and 5 (RR primary vs secondary 1.709 95% CI 1.599–1.897, p-value <0.001).

**Funding:** The research of DM is funded by COV20/00115 (Insituto de Salud Carlos III, Spain). The research of JP is funded by grants 2017SGR1049 (Agència de Gestió d'Ajuts Universitaris i de Recerca, Catalonia) and: PGC2018-098676-B-100 (Ministerio de Ciencia, Innovación y Universidades, Spain). The funders had no role in study design, data collection and analysis, decision to publish, or preparation of the manuscript.

**Competing interests:** The authors have declared that no competing interests exist.

## Conclusions

Safety measures adopted by schools were not enough to stop related-to-school transmission in students and could be improved. The safest way to keep schools open is to reduce community transmission down to a minimum.

## Introduction

After the declaration of the SARS-CoV-2 pandemic on the 11th of March 2020, schools closures were implemented by many countries as one of the key measures to control the initial phase of the pandemic [1–3]. Schools have been classically recognised as spreading places for infectious diseases, as they are usually confined and crowded spaces where close-contact between a high number of individuals occurs. As of September 2020, when many countries decided to reopen schools, little was known about SARS-CoV-2 transmission in school settings [4]. Several studies have now evaluated the impact of school closures on the evolution of the pandemic [2, 5, 6] or described outbreaks in schools and educational settings [7, 8]. Given the negative effects of the abrupt school closures in March 2020 on the health and well-being of children and teenagers [9, 10], which in many cases was combined with national lockdowns, several health and research groups issued recommendations for safe school reopening which focused on the use of safety measures in schools, protocols for reporting cases and isolating classrooms or schools and guidelines on when to open in-class education depending on the levels of community transmission or incidence [10–13].

In Catalonia (Spain), one of the hardest-hit regions by COVID-19 in Europe, primary and secondary schools opened on September 14th, when the incidence was around 180 PCR-confirmed cases per 100,000 in 14 days. Schools followed the Government of Catalonia [14] guidelines on safety measures (mask wearing for all children ≥6 years old, symptoms monitoring, hygiene and ventilation) and a procedure for reporting and managing PCR-confirmed positive SARS-CoV-2 infection cases. The organization of schools was designed around the concept of stable cohabitation groups in order to facilitate traceability of possible cases. The size of stable groups was recommended to be 20 in primary schools and 30 in secondary schools. When a positive case in one or more members of a school group was found, the whole group was considered a close contact and was isolated for 14 days (reduced to 10 days from October 1st) from the date of the last contact. Members of the cohabitation group, including the tutor, and other people having been in close contact with the case were recommended to be tested.

Even if COVID-19 among children is usually less severe and is more likely to be mild or asymptomatic than in adults [15] more data are needed to better understand how infectious children are and their role in transmission [16, 17]. Related-to-school transmission and whether it is only attributable to community transmission has been the subject of intense debate in Catalonia since September 2020. The goal of this study is to assess the impact of SARS-CoV-2 cases during weeks 1–2 of the academic year in school incidence in weeks 4–5, accounting for the community incidence in the area.

## Methods

### Study design, setting, participants

A retrospective cohort study was conducted among all students aged 3 to 18 years enrolled in all public schools in Catalonia during the first 5 weeks of the 2020/21 school year (September

14th-October 18th 2020). All data were obtained from publicly available databases from the Government of Catalonia's website. Public centres that have both primary and secondary education were excluded from the study, as cases are reported per center and not by age group. Only public schools were included, which represent approximately half of the schools in Catalonia.

## Data sources and computed variables

The Department of Education provides the daily number of PCR-confirmed cases among students, teachers and auxiliary personnel at every educational centre in Catalonia [18]. Data include the number of confined groups, students, teachers and auxiliary personnel, as well as the geographical coordinates of the centre, the education level (primary (ages 3 to 12), secondary (ages 12 to 18) or other) and whether it was public or private. The number of students of each centre was obtained from the database of enrolled students [19], which was only available for the 2019/20 school year. All these variables were matched by the centers' unique identifiers.

The main exposure variable was defined as the number of reported PCR-confirmed SARS-CoV-2 infections among students, teachers and auxiliary staff during weeks 1–2 of the school year (Sept 14-27th, 2020). The outcome of the study was the incidence of reported PCR-confirmed SARS-CoV-2 infections among students, teachers and auxiliary staff computed for weeks 4–5 of the school year (October 5-18th, 2020), calculated as the number of infections divided by the number of registered students. Positives of the first three weeks were excluded from the outcome denominator. Given that classmates are actively screened with a PCR when there is a case in a classroom, and that these are sometimes done with a few days delay, week 3 (Sept 28th—October 4th, 2020) was not included in the analysis, to avoid including PCR-positive contacts from weeks 1–2 in the outcome. We also assessed the effect of exposure to reported PCR-confirmed cases among students and adults (teachers and auxiliary staff) separately. Exposure to cases in weeks 1–2 was also considered as a dichotomous variable (none vs some case) to compare schools with at least one case and no cases in weeks 1–2.

Catalonia is divided into seven Sanitary Regions (SR) and each of these is divided into a number of subregions called Basic Health Areas (BHA). New PCR-confirmed positive cases for every BHA are publicly reported in a separate database [20]. The community-based cumulative incidence per 100,000 for the first two weeks was calculated by BHA as the reported number of PCR-positive cases in a BHA divided by its catchment population [21]. Finally, we assigned each school to its corresponding BHA and SR using the geographical shapefiles of each BHA [22] and the coordinates of each school.

## Analysis

Descriptive analysis of the population was performed using percentages in case of categorical variables, and median and the interquartile range (IQR) in case of continuous variables. School SARS-CoV-2 incidence over the different periods of the study (weeks 1–2, the excluded week 2 and weeks 4–5) for schools with and without cases during the first two weeks was graphically presented by COVID-19 Risk Level for reopening schools. The risk level was defined according to the Harvard Global Health Institute classification [11], which uses daily community incidence per 100,000 inhabitants: Green (<1 daily cases), Yellow (1–10), Orange (10–25) and Red (>25). The association between the exposure and the outcome was assessed with a univariate and multivariate multilevel Poisson regression with random effects at the Sanitary Region and BHA levels. The crude and adjusted Risk Ratios (RR) with the 95% Confidence Interval (CI) were obtained. The number of registered students per school excluding the cases from the

first three weeks were included in the model as offset [23]. Statistical analysis was performed using R version 4 [24] and library lme4 [25]. All code is available as S1 File.

### Ethics

This study used publicly available aggregated data.

## Results

This study includes a total of 2184 public schools (1625 primary and 558 secondary) which opened on September 14th and their 778,715 students (443,470 in primary and 335,245 in secondary schools). Characteristics of the study population during the two study periods are presented in Table 1. The age range was 3–12 for primary schools (although 73 schools (4.5%) had students aged 0–3). Most students of secondary schools are aged 12–18, but 190 secondary schools (34%) also have professional studies with students older than 18 (around 10% of enrolled students in secondary education). Schools reported 1152 PCR-confirmed SARS-CoV-2 infections during weeks 1–2 and 2412 during weeks 4–5. The median community incidence per 100,000 inhabitants in the 368 BHA during the first two weeks was 160.8 (IQR 105.7–223.5).

School incidence per 100,000 students of reported SARS-CoV-2 infections in weeks 1–2 was not significantly different in primary schools (139.1; CI 128.3–150.6) than in secondary schools (123.8; CI 112.1–136.3), whereas in weeks 4–5, the incidence in the secondary schools (364.5; 344.3–385.6) was higher than in primary schools (209.4; CI 196.1–223.3).

According to the Harvard Global Health Institute COVID-19 Risk levels for reopening, during the first two weeks of the school year, there were 29 centres (26 primary and 3 secondary) in green areas, 1009 in yellow areas (758 primary and 251 secondary), 945 in orange areas (684 primary and 261 secondary) and 200 in red areas (157 and 43 respectively).

Fig 1 shows weekly school incidence during the different study periods (weeks 1–2 and weeks 4–5) stratified by the community incidence level and classifying schools by whether they had cases during the weeks 1–2 or not. We excluded schools in Green areas from the plot because they reported only one case in each of the periods. Incidence in Yellow, Orange and Red areas was significantly higher in weeks 4–5 in centres with at least one reported positive case in the weeks 1–2 (incidence in person-weeks per 100,000: Yellow areas 112.8 (CI 97.1–130.2), Orange areas 198.7 (CI 183.4–214.8) and Red areas 396.6 (CI 349.1–448.9) than in centres which reported no cases in weeks 1–2 (Yellow areas 85.1 (CI 77.4–93.4), Orange areas 125.3 (CI 115.0–136.2) and Red areas 188.4 (CI 150.3–233.2)).

**Table 1. Characteristics of the study population by type of centre and period.**

| | Weeks 1–2 | | Weeks 4–5 | |
|---|---|---|---|---|
| | **Primary Schools** | **Secondary Schools** | **Primary Schools** | **Secondary Schools** |
| Total number of new PCR-positives (students+staff) | 684 | 468 | 1105 | 1307 |
| Number of PCR-positives among students | 617 | 415 | 926 | 1219 |
| Number and % of centres with PCR-positives | | | | |
| 0 cases | 1243 (76.5%) | 322 (57.7%) | 1102 (67.8%) | 174 (31.2%) |
| 1 case | 224 (13.8%) | 126 (22.6%) | 274 (16.9%) | 140 (25.1%) |
| 2 cases | 85 (5.2%) | 52 (9.3%) | 127 (7.8%) | 76 (13.6%) |
| 3 cases | 37 (2.3%) | 35 (6.3%) | 44 (2.7%) | 56 (10.0%) |
| 4 or more | 36 (2.2%) | 23 (4.1%) | 75 (4.8%) | 112 (20.1%) |

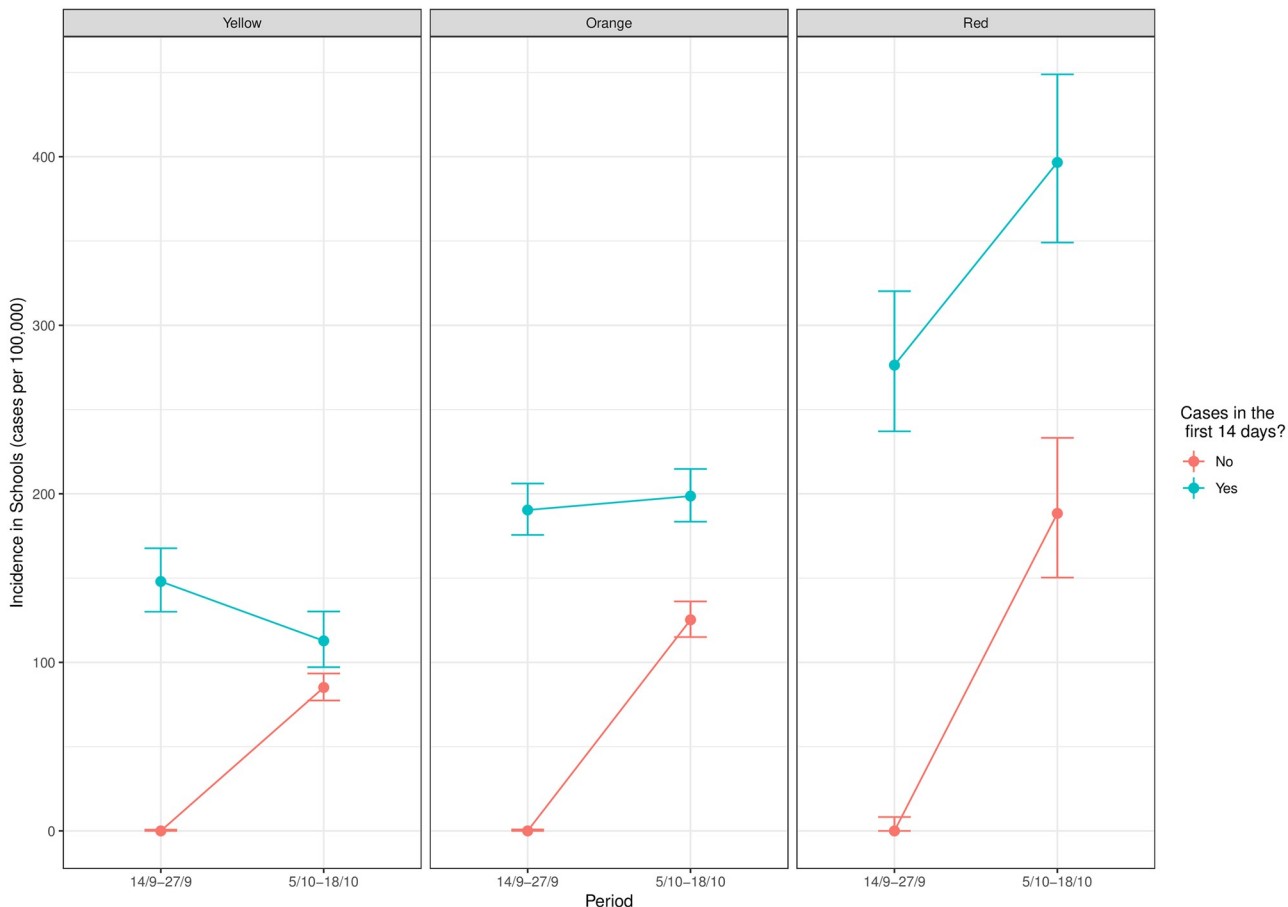

**Fig 1. Weekly SARS-CoV-2 school incidence (cases per 100,000 students in 14 days) for weeks 1–2 and weeks 4–5 of the academic year in all public schools which had and did not have cases in weeks 1–2, stratified by the community incidence levels in the corresponding Basic Health Areas, as defined by the Harvard Global Health Institute [11] COVID-19 Risk Levels for reopening schools by daily incidence per 100,000 inhabitants (Yellow (1–10), Orange (10–25) and Red (>25)).**

During the study period, all public schools in Catalonia remained open. However, at the end of the first two weeks, 5,856 primary school students (1.3%; CI 1.2–1.4) and 5,475 secondary school students (1.6%; CI 1.5–1.7) were isolated or quarantined, increasing to 9,016 (2.03% CI 1.9–2.1) in primary schools and 9,656 (2.9% CI 2.8–3.0) in secondary schools at the beginning of weeks 4 and 5. On the last study day, the number of isolated/quarantined students was 8,616 (1.9% CI 1.9–2.0) for primary and 11,659 (3.5% CI 3.4–3.6) for secondary schools.

The multilevel analysis with random effects at the Sanitary Region and BHA levels (Table 2) showed a significant association between the total number of PCR-positives in a centre in weeks 1–2 and the school SARS-CoV-2 infection incidence in weeks 4–5 (RR 1.074, CI 1.044–1.105). The community BHA incidence also showed a significant association with the school SARS-CoV-2 infection incidence in weeks 4 and 5 (RR 1.002, CI 1.002–1.003). When disaggregating exposure by PCR-positive students or staff, only exposure to PCR-positive students was significant (students RR 1.083, CI 1.051–1.115; staff RR 0.953, CI 0.835–1.086). Since the interaction between the total number of positives in a centre in weeks 1–2 and the type of school was not significant (RR 0.96, CI 0.91–1.01), a combined RR for primary and secondary schools is presented. When considering exposure to cases in weeks 1–2 as dichotomous variables, the

**Table 2. Univariate and multivariate multilevel Poisson regression analysis of potential determinants for incidence of SARS-CoV-2 cases in public schools in Catalonia in weeks 4–5.**

| Risk Ratio of SARS-CoV-2 infection incidence in weeks 4–5 | | |
| --- | --- | --- |
| | Univariate | Adjusted |
| Reported SARS-CoV-2 infections among students and staff in weeks 1–2 | 1.175*** (1.154–1.196) | 1.074*** (1.044–1.105) |
| Community incidence in the Basic Health Area (cases per 100,000 inhabitants) in weeks 1–2 | 1.002*** (1.002–1.003) | 1.002*** (1.002–1.003) |
| Type of School: Primary (as reference category) versus Secondary | 1.741*** (1.599–1.897) | 1.709*** (1.564–1.867) |

*** p<0.001.

significance between the variables in Table 2 remained the same (exposure to cases in weeks 1 and 2 (RR 1.281, CI 1.159–1.416).

## Discussion

In this study, we describe SARS-CoV-2 reported cases in public school students in the first five weeks of the 2020–21 academic year in the public schools of Catalonia. We found a significant association between the number of reported SARS-CoV-2 infections among students and staff in weeks 1–2 and the incidence among students in weeks 4–5. In the models presented, for every PCR-positive case reported in weeks 1–2, the incidence in weeks 4–5 increased around 7%. This increased risk becomes more relevant if we take into account that over 9% of primary schools and 20% of secondary schools had 2 or more cases during the first two weeks. This suggests that there may be related-to-school transmission, which includes not only transmission occurring within the school but also that related to social interactions around school attendance, when getting to or leaving the school.

Related-to-school transmission is part of community transmission and different studies have tried to measure the contribution of school attendance to community transmission [6, 26]. For instance, school closure was associated with a decrease in COVID-19 incidence in the USA [5]. Another modelling study [27] suggested that in the absence of large-scale testing, contact tracing, and isolation, full time reopening of schools may have contributed to a second wave in the UK. A study of lower-secondary schools in Sweden suggested a minor impact on the overall spread [28]. The contribution of in-school transmission in the community is still subject to much debate [29, 30].

SARS-CoV-2 is usually underdiagnosed among children [31], probably because of the large proportion of asymptomatic patients compared to adults. A large sero-prevalence survey in Spain showed that children had been less affected during the first wave of the pandemic than adults [32], probably because they were better protected due to school closures and lockdowns. However, from June 22nd to late November 2020, all age groups sero-converted in the same proportion [33]. A meta-analysis of studies until July 28th 2020 suggested that the susceptibility of children to SARS-CoV-2 infection was lower than in adults, although the role of children and adolescents in the transmission was unclear [15]. Our study sheds some light on the role of children in transmission since the results suggest transmission among students. This is aligned with studies that showed that children in quarantined households are equally likely to become infected by SARS-CoV-2 as adults [34]. Also, the transmission of SARS-CoV-2 among household members can be originated from either children or adults [35], although different studies suggest the same [36] or lower [37] infectivity in children. A large contact-tracing study in India observed an increased risk of transmission among children and young adults

[38] and a large cohort study in the UK showed an increased risk of reported SARS-CoV-2 infection and COVID-19 hospitalisations among adults living with children in the second wave [39]. In the context of our research, the Epidemiological Surveillance Network of Catalonia's report corresponding to the study period, reported 197 active outbreaks (defined as three or more confirmed cases with epidemiological link) in educational settings, which comprised 1,091 infected people. This represented 14% of all outbreaks reported [40].

The incidence in the community in weeks 1–2 plays a role in the school incidence in weeks 4–5. Adjusting for the rest of the variables, for every 100 cases increase in the BHA incidence (measured as cases per 100,000 inhabitants in weeks 1–2), school incidence increased by around 20%. Even if we do not analyze the increase in incidence in Fig 1, it is reasonable to expect that the relative growth in incidence in schools which did not report cases in weeks 1–2 (plot in red) is higher with respect to schools which do report cases (plot in green). Indeed, schools which report cases in weeks 1–2 applied contention measures (isolation and quarantines) and, if measures were enough, one would expect that incidence is not higher than in schools which do not report cases. It has also been reported [41] that the number of confined school groups in the first two weeks of the academic year was associated with the community incidence of the Sanitary Region in the same period, which is consistent with our results.

Incidence in secondary schools was around 70% higher than in primary schools in weeks 4–5. The variables included in this study do not allow us to explain this difference that was not detected in weeks 1–2. A possible explanation is the number of students per class, higher in secondary. Besides, social interaction and physical contact differs by age, both within schools and around school attendance, and going to school might also include other activities which begin with the school year that increase the infection risk, like the use of public transportation, the beginning of the extracurricular activities or jobs and internships in companies. More research is needed to understand these differences. Enforcing contact-tracing and to take into account students' voices could help to detect, understand and prevent these contagions.

More than 30,000 students had been infected by SARS-CoV-2 in Catalonia by the end of December 2020. Although symptoms in children are usually mild, its long term effects are largely unknown. The impact of high community transmission is also shown in the number of students that have been quarantined during the term: as of December 13th around 300,000 students (about 20% of all students) have been quarantined. School closures, isolations and quarantines have negative effects in children's health and education, their parents lives and professional activity, especially in the increase of inequities [42], as many other aspects of the pandemic [43]. A study in Italy about school opening in the same period but with lower incidence [44] showed that, after a month of opening, 1,305 cases were reported in Italian schools which caused the closure of 192 schools (around 1% of the total number of schools), to avoid further secondary cases, isolations and quarantines. In comparison, no schools in Catalonia were closed during the study period. The original protocol [14], however, suggested that schools should be partially or totally closed when two or more cases were detected. Different guidelines state that, when there is widespread community transmission, school closures may be part of public health and social measures to prevent the spread of SARS-CoV-2 virus [10] and that the administration should invest in distance learning and guarantee basic services, both resources and support, to ensure the right to education and well-being of all children [11]. It would be interesting to describe the social and economic impact of quarantines and isolations linked to school cases. Planning school holidays to be coordinated if necessary with wider population restrictions could help reduce transmission alongside with a lesser economic, educational and social impact. Prediction models which take into account socioeconomic dimension may be useful in this planning.

Apart from sharing classrooms, students do usually share spaces and, although we were not able to consider these events separately, it may be reasonable to increase protective measures in these settings. Besides it could be advisable to reduce the number of students per classroom (for instance, allowing online learning and providing the resources for those children and families who prefer it), to perform a PCR to close contacts after the incubation period before returning to school and that people at medical risk (students, staff and their families) should be given the chance, the resources and support to switch to online education or teaching, to preserve their health. More research is needed on the safety and organization measures which help to avoid transmission. The whole educational community should be involved in the changes and actions to improve safety.

A limitation of this study is that data were aggregated at the school level and it was not possible to know if different SARS-CoV-2 infection cases shared the same classroom, school canteen, bus or extracurricular activities or which was their age group. This does not allow the assessment of the transmission routes inside the schools, neither to distinguish between pre-school (3–5 years old) and primary school (6–12 years old), which had differences in the mask use protocol. We were not able to quantify the compliance of the different schools with the protective measures (mask wearing, frequent hand washing and ventilation) and isolations. Some factors might underestimate the association between PCR-positives in weeks 1–2 and increased incidence in weeks 4–5. First of all, after a PCR-positive was detected, students and teachers who were considered close contacts were tested by PCR, finding some new PCR-positives, including some who could be index cases. To account for this possible effect, we did not include the third week of the academic year in the analysis. Secondly, students were not routinely tested at the end of the quarantine, so infected asymptomatic or very mild cases upon return might have gone undetected. Exposure to infected teachers and auxiliary staff was not significant, probably because the sample size was too small. A strength of this study is the large number of schools and students, which represent roughly half of the schools in Catalonia. Also, these schools depend directly from the Department of Education, making the reporting and adherence to the protocols more homogeneous.

Safety measures seem insufficient to prevent transmission in schools which had reported cases in the previous weeks. The safest way to reopen schools, to protect the health of students, staff and their families, seems to be to improve case and contacts management and reduce transmission in the community to minimal levels [4], which will be easier when vaccines are available.

## Supporting information

**S1 File. Computer code.** Contains a R file including the analysis.
(RMD)

**S2 File. Dataset.**
(CSV)

## Acknowledgments

We are grateful to the Department of Health and the Department of Education of the Government of Catalonia for making data publicly available.

## Author Contributions

**Conceptualization:** Anna Llupià, Joaquim Puig.

**Data curation:** Joaquim Puig.

**Formal analysis:** Anna Llupià, Alícia Borràs-Santos, Mireia Utzet, Joaquim Puig.

**Investigation:** Anna Llupià, Alícia Borràs-Santos, Mireia Utzet, Joaquim Puig.

**Methodology:** Anna Llupià, Alícia Borràs-Santos, Caterina Guinovart, Mireia Utzet, David Moriña, Joaquim Puig.

**Software:** Alícia Borràs-Santos, Caterina Guinovart, Mireia Utzet, David Moriña, Joaquim Puig.

**Supervision:** Anna Llupià, Alícia Borràs-Santos, Caterina Guinovart, Mireia Utzet, David Moriña, Joaquim Puig.

**Validation:** Anna Llupià, Alícia Borràs-Santos, Caterina Guinovart, Mireia Utzet, David Moriña, Joaquim Puig.

**Visualization:** Joaquim Puig.

**Writing – original draft:** Anna Llupià, Alícia Borràs-Santos, Joaquim Puig.

**Writing – review & editing:** Anna Llupià, Alícia Borràs-Santos, Caterina Guinovart, Mireia Utzet, David Moriña, Joaquim Puig.

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
