## [Decision Letter · Decision Letter 0]

2 Mar 2021

PONE-D-21-02405

SARS-CoV-2 transmission in students of public schools of Catalonia (Spain) after a month of reopening

PLOS ONE

Dear Dr. Llupià,

Thank you for submitting your manuscript to PLOS ONE. After careful consideration, we feel that it has merit but does not fully meet PLOS ONE’s publication criteria as it currently stands. Therefore, we invite you to submit a revised version of the manuscript that addresses the points raised during the review process.

We look forward to receiving your revised manuscript.

Kind regards,

Beatriz da Costa Thomé, MD, MPH, PhD

Academic Editor

PLOS ONE

Journal Requirements:

Additional Editor Comments:

Valuable article that discusses a relevant aspect of Covid transmission that relates to the contribution of school. A major strength is making use of publicly available aggregated data. The data however has important limitations that should also limit the strength of the conclusions. Furthermore, the discussion misses in my opinion other references in which school transmission of course reflected community transmission (one of the findings of this article), but didn't contribute so importantly to overall transmission. These should be added to the discussion.

Reviewers' comments:

Reviewer's Responses to Questions

**Comments to the Author**

1. Is the manuscript technically sound, and do the data support the conclusions?

Reviewer #1: Yes

Reviewer #2: Yes

Reviewer #3: Partly

2. Has the statistical analysis been performed appropriately and rigorously? 

Reviewer #1: Yes

Reviewer #2: Yes

Reviewer #3: No

3. Have the authors made all data underlying the findings in their manuscript fully available?

Reviewer #1: Yes

Reviewer #2: Yes

Reviewer #3: Yes

4. Is the manuscript presented in an intelligible fashion and written in standard English?

Reviewer #1: Yes

Reviewer #2: Yes

Reviewer #3: Yes

5. Review Comments to the Author

Reviewer #1: The manuscript by Anna Llupià et al. investigates the impact of re-opening schools in the transmission of SARS-CoV-2 through a retrospective cohort using available public data. The manuscript's strengths are the population level of the data. They found an association between the number of cases in the first two weeks of re-opening with the increased incidence of cases 2-3 weeks later, particularly in secondary schools. Although these findings are significant and obtained via adequate statistical analysis, some conclusions and inferences need to be more suitable when presenting and discussing the results. The existence and the impact of within school transmission can not be inferred with the data available and with this study design. When adjusted by community incidence (BHA), the RR is significant but very mild (RR 1.002). Also, the increase in incidence at week 4-5 was higher when the schools had zero cases in the first two weeks compared to those with at least one case (Figure 1). It does not reject the study and its findings, but the impact of within school transmission should be cautiously stated and discussed better.

Minor comments:

- When discussing the higher incidence in secondary schools, the authors mentioned several behaviours and social reasons, but they do not mention the higher number of students per class (20 vs 30 students, as mentioned in the introduction).

- The methods section in the abstract is a bit confusing. I suggest rewriting.

- Page 2 / Line 9 – wrong capital letter in To; two "s" in sSARS-CoV-2

- Page 10 / Line 18 - "may have contributed"

- Page 10 / Line 21 – no comma after Sapin

- Page 11 / Line 19 – ineffective

- Page 11 / Lines 25-26 – confusing sentence; suggest rewriting

- Page 12 / Line 2 – SARS-COV-2

- Page 13 – paragraph starting in line 5 is too long and hard to read.

Reviewer #2: Please review line 9 page 4

The article is relevant to this moment of school reopening demand, data presents the impact of reopening to the number of cases and the risk of it. If possible, it would be interesting to consider impact of vaccination in these process.

Reviewer #3: Relevant study, describes appropriately the SARS-CoV-2 transmission in public school students in the first five

4 weeks of the 2020-21 academic year in the public schools of Catalonia-Spain, using available aggregated data.

Simple analysis and it is not described in sufficient detail. The study presents the results of original research, but the analysis must be describe in sufficient detail. Also, the conclusions are not presented in an appropriate fashion.

The authors must highlight the limitation of this study related to data were aggregated at the school level and it was not possible to know if different SARS-CoV2 infection cases shared the same place or extracurricular activities or which was their age group.

6. PLOS authors have the option to publish the peer review history of their article (what does this mean?). If published, this will include your full peer review and any attached files.

Reviewer #1: No

Reviewer #2: **Yes: **Mariana Cabral Schveitzer

Reviewer #3: No

---

## [Author Response · Author response to Decision Letter 0]

22 Mar 2021

(a docx version of this letter is available in the 'response to reviewers' file)

Comments to the Author. Reviewer #1

- The existence and the impact of within school transmission can not be inferred with the data available and with this study design. When adjusted by community incidence (BHA), the RR is significant but very mild (RR 1.002). 

The RR of 1.002 refers to the risk of community incidence in the BHA, in terms of cases per 100,000. In lines 10-13 of page 11 (original submission) we state that “The incidence in the community in the first two weeks plays an important role in the school incidence in weeks 4 and 5. Adjusting for the rest of the variables, for every 100 cases increase in the BHA incidence (measured as cases per 100,000 inhabitants in the first two weeks), school incidence increased by around 20%.”. Note that incidence was around 100-200 cases per 100,000 in weeks 1 & 2. 

We agree that the impact of within school transmission cannot be inferred from the data available and with this study design. To avoid confusion, we have removed the following sentence from the discussion: “When this transmission is high, our findings imply that related-to-school infections do commonly occur and get probably transmitted to households through children and teenagers” (lines 22-24, page 12 submitted version).

- Also, the increase in incidence at week 4-5 was higher when the schools had zero cases in the first two weeks compared to those with at least one case (Figure 1). 

In our study we do not evaluate the increase in incidence in weeks 4-5 with respect weeks 1 and 2 and we included Figure 1 to illustrate that incidence in weeks 4-5was higher in schools which had reported cases in weeks 1-2 with respect to those which did not report any, and that this holds disaggregating by community incidence in the area. We agree that the writing of one sentence may induce confusion and we have removed it (lines 13-16 in page 11, submitted version). 

Even if we do not analyze the increase in incidence in Figure 1, it is reasonable to expect that the relative growth in incidence in schools which did not report cases in weeks 1-2 (plot in red) is higher with respect to schools which do report cases (plot in green). Indeed, schools which report cases in weeks 1-2 applied contention measures (isolation and quarantines) and, if measures were enough, one would expect that incidence is not higher than in schools which do not report cases. A conclusion of our study is that these contention measures are not enough to stop transmission and the incidence in weeks 4-5 is still higher in schools which report cases in weeks 1-2. The new version of the discussion has incorporated these observations.

- It does not reject the study and its findings, but the impact of within school transmission should be cautiously stated and discussed better.

Discussion and conclusions have been rewritten in order to be more cautious with the conclusions of the impact of in-school transmission. 

Minor comments.

- In the discussion we have included the following sentence to account for the possibility that higher classroom size may explain the increased risk in secondary schools: “A possible explanation is the number of students per class, higher in secondary. “ (page 11, lines 20-21, submitted version). 

- Page 2 / Line 9 – wrong capital letter in To; two "s" in sSARS-CoV-2: Changed

- Page 10 / Line 18 - "may have contributed". Changed.

- Page 10 / Line 21 – no comma after Spain. Changed.

- Page 11 / Line 19 – ineffective. Changes from “infective” to “infectious”. 

- Page 11 / Lines 25-26 – confusing sentence; suggest rewriting. We have changed the writing and we now think that it is clearer together with an extra sentence in the same paragraph (see version with tracked changes). 

- Page 12 / Line 2 – SARS-COV-2. Changed

- Page 13 – paragraph starting in line 5 is too long and hard to read. It has been reorganized. 

Comments to the Author. Reviewer #2

- Please review line 9 page 4: We have corrected the two misprints.

- If possible, it would be interesting to consider impact of vaccination in these process. We have updated the sentence in the last paragraph adding “which will be easier when vaccines are available.” Note that during the study period (ending in mid-October 2020, vaccination had not yet been rolled out).

Comments to the Author. Reviewer #3

Simple analysis and it is not described in sufficient detail. The study presents the results of original research, but the analysis must be describe in sufficient detail. 

We have revised the methods section to clarify some points of the analysis description. We used a multilevel Poisson regression with random effects at the Sanitary Region and BHA levels, with an offset at the number of students minus number of registered students per school excluding the cases from the first two weeks. All the analysis and datasets are available as supplementary files. 

Also, the conclusions are not presented in an appropriate fashion.

Conclusions are now like this: “Safety measures seem insufficient to prevent cases in schools which had reported cases in the previous weeks. The safest way to reopen schools, to protect the health of students, staff and their families seems to improve case and contacts management and reduce transmission in the community to minimal levels [4], which will be easier when vaccines are available.”

The conclusions of the abstract have been rewritten accordingly. 

The authors must highlight the limitation of this study related to data were aggregated at the school level and it was not possible to know if different SARS-CoV2 infection cases shared the same place or extracurricular activities or which was their age group.

We agree that it is important to highlight the limitation of the study related to the aggregation of data at school level and not classroom level and the study design. To emphasize this, we have removed the sentence “When this transmission is high, our findings imply that related-to-school infections do commonly occur and get probably transmitted to households through children and teenagers” and we have rewritten the 

paragraph of limitations (page 12, lines 1-18 in the original submission).

---

## [Decision Letter · Decision Letter 1]

29 Apr 2021

SARS-CoV-2 transmission in students of public schools of Catalonia (Spain) after a month of reopening

PONE-D-21-02405R1

Dear Dr. Llupià,

We’re pleased to inform you that your manuscript has been judged scientifically suitable for publication and will be formally accepted for publication once it meets all outstanding technical requirements.

Kind regards,

Beatriz Thomé, MD, MPH, PhD

Academic Editor

PLOS ONE

Additional Editor Comments (optional):

Thanks for responding to the comments made during the first review of the manuscript. A few remaining (minor) suggestions are shared for the authors consideration as they proceed with the final version of the article.

Reviewers' comments:

Reviewer's Responses to Questions

**Comments to the Author**

1. If the authors have adequately addressed your comments raised in a previous round of review and you feel that this manuscript is now acceptable for publication, you may indicate that here to bypass the “Comments to the Author” section, enter your conflict of interest statement in the “Confidential to Editor” section, and submit your "Accept" recommendation.

Reviewer #1: All comments have been addressed

Reviewer #2: All comments have been addressed

Reviewer #3: All comments have been addressed

2. Is the manuscript technically sound, and do the data support the conclusions?

Reviewer #1: Yes

Reviewer #2: Yes

Reviewer #3: Yes

3. Has the statistical analysis been performed appropriately and rigorously? 

Reviewer #1: Yes

Reviewer #2: Yes

Reviewer #3: Yes

4. Have the authors made all data underlying the findings in their manuscript fully available?

Reviewer #1: Yes

Reviewer #2: Yes

Reviewer #3: Yes

5. Is the manuscript presented in an intelligible fashion and written in standard English?

Reviewer #1: Yes

Reviewer #2: Yes

Reviewer #3: Yes

6. Review Comments to the Author

Reviewer #1: After the first review, the authors provided accurate responses, making the manuscript neater and more precise after the revision. I do not suggest any significant corrections.

Minor comments:

Line 83, 243, 244 – I suggest adding the year (2020) to make it easier for future readers.

Title and first line in Table 2 and Line 281 – Correct SARS-CoV2 to SARS-CoV-2.

Line 221 – Delete “describe”.

Line 253 – I would clarify "COVID-19 outcomes" (worst outcome if children in the household?).

Line 319 – correct “all,after”.

Reviewer #2: The update was very helpful. There are considerations today regarding the PCR negative results as the only criteria to return to activities. A possible approach would be to consider clinical observation and patient lack of symptoms. How is this discussion in Catalonia?

Reviewer #3: (No Response)

7. PLOS authors have the option to publish the peer review history of their article (what does this mean?). If published, this will include your full peer review and any attached files.

Reviewer #1: **Yes: **Vinicius Adriano Vieira

Reviewer #2: No

Reviewer #3: No

---

## [Editor Report · Acceptance letter]

10 May 2021

PONE-D-21-02405R1 

SARS-CoV-2 transmission in students of public schools of Catalonia (Spain) after a month of reopening 

Dear Dr. Llupià:

I'm pleased to inform you that your manuscript has been deemed suitable for publication in PLOS ONE. Congratulations! Your manuscript is now with our production department. 

Kind regards, 

on behalf of

Dr. Beatriz Thomé 

Academic Editor

PLOS ONE